# Complete Genome Sequencing of Tick-Borne Encephalitis Virus Directly from Clinical Samples: Comparison of Shotgun Metagenomic and Targeted Amplicon-Based Sequencing

**DOI:** 10.3390/v14061267

**Published:** 2022-06-10

**Authors:** Samo Zakotnik, Nataša Knap, Petra Bogovič, Tomaž Mark Zorec, Mario Poljak, Franc Strle, Tatjana Avšič-Županc, Miša Korva

**Affiliations:** 1Institute of Microbiology and Immunology, Faculty of Medicine, University of Ljubljana, SI-1000 Ljubljana, Slovenia; samo.zakotnik@mf.uni-lj.si (S.Z.); natasa.knap@mf.uni-lj.si (N.K.); tomaz-mark.zorec@mf.uni-lj.si (T.M.Z.); mario.poljak@mf.uni-lj.si (M.P.); tatjana.avsic@mf.uni-lj.si (T.A.-Ž.); 2Department of Infectious Diseases, Ljubljana University Medical Center, SI-1000 Ljubljana, Slovenia; petra.bogovic@kclj.si (P.B.); franc.strle@kclj.si (F.S.)

**Keywords:** tick-borne encephalitis virus, whole genome sequencing, amplicons, metagenomics, next-generation sequencing (NGS), clinical samples

## Abstract

The clinical presentation of tick-borne encephalitis virus (TBEV) infection varies from asymptomatic to severe meningoencephalitis or meningoencephalomyelitis. The TBEV subtype has been suggested as one of the most important risk factors for disease severity, but TBEV genetic characterization is difficult. Infection is usually diagnosed in the post-viremic phase, and so relevant clinical samples of TBEV are extremely rare and, when present, are associated with low viral loads. To date, only two complete TBEV genomes sequenced directly from patient clinical samples are publicly available. The aim of this study was to develop novel protocols for the direct sequencing of the TBEV genome, enabling studies of viral genetic determinants that influence disease severity. We developed a novel oligonucleotide primer scheme for amplification of the complete TBEV genome. The primer set was tested on 21 clinical samples with various viral loads and collected over a 15-year period using the two most common sequencing platforms. The amplicon-based strategy was compared to direct shotgun sequencing. Using the novel primer set, we successfully obtained nearly complete TBEV genomes (>90% of genome) from all clinical samples, including those with extremely low viral loads. Comparison of consensus sequences of the TBEV genome generated using the novel amplicon-based strategy and shotgun sequencing showed no difference. We conclude that the novel primer set is a powerful tool for future studies on genetic determinants of TBEV that influence disease severity and will lead to a better understanding of TBE pathogenesis.

## 1. Introduction

Tick-borne encephalitis (TBE) is one of the most important viral infections of the human central nervous system transmitted by arthropods. It is caused by the tick-borne encephalitis virus (TBEV), which is endemic throughout Europe (27 countries) and parts of Asia (at least four countries). TBEV is phylogenetically divided into three main subtypes—European (TBEV-Eu), Siberian (TBEV-Sib), and Far Eastern (TBEV-FE) [1]—as well as two subtype lineages—“178-79” and “886-84” (Baikal TBEV subtype TBEV-Bkl) [2]—and a potential Himalayan subtype (TBEV-Him) [3]. The TBEV-Eu subtype is common in Europe, whereas TBEV-Sib and TBEV-FE are mainly found in Asia. There are geographical areas where all three main subtypes coexist (the Baltic states, Siberia, Ukraine, and European parts of Russia) [4]. In central Europe, TBE is a seasonal disease, with most cases occurring from April to November [5]. The most common mode of transmission is the bite of an infected tick, but other routes of transmission are also possible. Approximately 1% of all infections are caused by the consumption of unpasteurized dairy products [6] or, less frequently, by solid organ transplantation [7] or laboratory infection [8]. In nature, TBEV circulates between an arthropod vector and its hosts. The most important tick species for transmission of TBEV to humans are *Ixodes ricinus* and *I. persulcatus* [9,10].

TBEV has a spherical virion about 50 nm in diameter that encapsulates the viral genome in the form of single-stranded positive-sense RNA. The viral genome is approximately 10 kb long and encodes a single polyprotein that is cleaved cotranslationally and post-translationally by host and viral proteases. The cleavage results in three structural proteins (C, PrM, and E) and seven non-structural proteins (NS2A, NS2B, NS3, NS4A, NS4B, and NS5) [11]. Human infections with TBEV have highly variable courses, including asymptomatic infection, febrile illness without central nervous system (CNS) involvement, or various forms of CNS inflammation (TBE), ranging from a mild uneventful course to a fatal outcome. In patients with TBE, older age is a major risk factor for severe disease [12,13], and other host factors that influence the disease course and outcome are still largely unknown. On the other hand, TBEV subtypes have been associated with various disease courses and mortality rates. TBE caused by TBEV-Eu is usually biphasic, with case fatality rates ranging from 0.5 to 2.0% [14]. The first, viremic phase of TBEV-Eu infection begins about 8 days after a tick bite, with fatigue, headache, and moderate fever, lasting for approximately a week. Neurological symptoms appear in the second phase of the disease, which usually follows a week-long improvement. In addition to acute CNS inflammation, 1.0 to 1.7% of patients infected with the TBEV-Sib subtype could develop a chronic course of the disease, with a fatality rate of 6.0 to 8.0% [13]. The TBEV-FE subtype is associated with the most severe disease, which typically has a monophasic course and a 5.0 to 20.0% fatality rate. It is also associated with the hemorrhagic form of the disease [14,15].

Slovenia, where 2454 TBE cases were recorded between 2008 and 2021, has one of the highest TBE incidences in Europe, with 8.1 to 18.6 cases per 100,000 population [16]. Approximately 75% of patients have a typical biphasic course of the disease; in the second phase, 50% of patients develop meningitis, 40% meningoencephalitis, and 10% meningoencephalomyelitis [12]. Because the only confirmed TBEV subtype in Slovenia is TBEV-Eu [17], different courses, severity, and outcomes of the disease are not related to other TBEV subtypes. The hypothesis on the relationship between the TBEV genome and disease course and outcome is intriguing but difficult to explore because the virus can only be detected in patients in the viremic phase, when, due to mild symptoms, patients rarely seek medical help and, due to nonspecific presentation, they are only exceptionally diagnosed with TBEV infection [18,19]. In addition, clinical isolates of TBEV are scant, and the amount of genetic material required for sequencing requires multiple passages of the virus in cell lines or in laboratory animals [20,21]. Because the virus could acquire novel mutations and/or lose genetic material during passage(s) in cell lines or animal hosts, a spurious correlation between genomic mutation and disease progression can be projected [22].

Recently, second-generation sequencing (also known as next-generation sequencing, NGS) has become a widely available research tool in many laboratories around the world. Affordable and less-demanding third-generation sequencing with Oxford Nanopore Technologies has removed the last technological barriers, and consequently, sequencing is no longer a bottleneck in research [23,24,25,26]. The unbiased nature of NGS and the ability to sequence a genome directly from clinical samples without preceding viral isolation or culture (i.e., metagenomics) has already been used successfully in both research and diagnostic settings [7,27]. However, the sequencing of viral genomes directly from clinical samples remains a challenge for archived specimens and/or samples with low viral loads. Usually, pre-amplification of the viral genome is required to overcome these challenges. Recently, several amplicon-based methods have been developed for the sequencing of viruses, such as Ebola virus [28], Zika virus [29], West Nile virus [30], TBEV [31], and SARS-CoV-2 (Artic Network). However, the previously described amplicon-based sequencing approach for TBEV is based on large overlapping amplicons (approximately 2 kb) and is designed for samples with high viral loads [31].

The aim of this study was to develop and validate novel protocols for sequencing complete TBEV genomes directly from clinical samples, such as blood and urine, without the need for preceding virus isolation and additional passage in in-vitro systems. Because a high viral load is rarely present in the whole-blood specimens of TBEV-infected patients [18], we designed a novel set of primer pairs with short amplicons in a range of 450 bp using the “jackhammering” strategy, which, unlike the long-amplicon approach [29], is suitable for samples with low viral loads and partially degraded samples [32]. In addition, we investigated shotgun metagenomic sequencing, which did not require additional specific amplification of the TBEV genome, and we compared the data obtained between second- and third-generation sequencing.

## 2. Materials and Methods

### 2.1. Patients and Samples

The specimens were obtained as part of a prospective study of the etiology of febrile illness after a tick bite or represented remnants of specimens collected as part of the routine diagnostic workup of patients with a febrile illness that later developed CNS involvement (TBE). Twenty-one patients were included in the study, all of whom were treated at the Department of Infectious Diseases, Ljubljana University Medical Center, Slovenia, between 2003 and 2020. The presence of TBEV RNA was confirmed by real-time RT-PCR in the patients’ serum or EDTA blood sample [33]. After diagnosis, the sample residues were stored at −80 °C until further processing.

### 2.2. TBE Infection Case Definition

TBE was defined as clinical symptoms of meningitis or meningoencephalitis, elevated CSF leukocyte counts (>5 × 106 cells/L), and the presence of TBE IgM and IgG antibodies in serum or IgG conversion in paired serum samples. The first phase of TBE was classified as a febrile illness that, after clinical improvement with a duration of up to 3 weeks, was followed by neurologic involvement fulfilling the criteria for TBE.

### 2.3. Ethics Considerations

The study was conducted in accordance with the principles of the Declaration of Helsinki and the Oviedo Convention on Human Rights and Biomedicine and was approved by the National Medical Ethics Committee of Slovenia (no. 152/06/13, no. 178/02/13, and no. 37/12/13). The patients whose specimens were collected as part of the study on the etiology of febrile illnesses after a tick bite signed an informed consent form that included the use of the collected specimens for further studies. The ethics committee waived the requirement for written informed consent for patients in whom remnants of routinely collected serum samples for diagnostic workup were used.

### 2.4. Available TBEV Genome Sequences

As of 12 April 2022, in total, 280 genomes of TBEV (sequences longer than 10 kbp) have been published in the NCBI database NT (https://www.ncbi.nlm.nih.gov/nuccore; accessed on 12 April 2022). Of these, 123 were sequenced from ticks, 106 from humans, 32 from small mammals, 6 from birds, 3 from mosquitoes, and 1 each from a goat, sheep, and monkey. Seven sequences have unknown sources. Only 2 of the 106 human-derived TBEV genomes were sequenced directly from clinical samples; all others were first processed via cell culture or inoculated into laboratory animals.

### 2.5. Viral Load

Quantitative real-time reverse transcription PCR (RT-PCR) was performed using TaqMan Fast Virus 1-Step Master Mix (Thermo Fisher Scientific Inc., Waltham, MA, USA) as described previously [33].

### 2.6. Design of Oligonucleotide Primer Pairs for Amplicon Amplification of TBEV Genome

To design oligonucleotide primer pairs for amplification of the complete TBEV genome, we used the Online Primal Scheme Primer Designer software [29]. As a reference genome, we used the genome sequence of TBEV strain Ljubljana I, which is deposited in the GenBank database under the accession number JQ654701 [17]. The designed primer sequences can be found in Appendix A.

### 2.7. PCR Amplification and Optimization of Primer Pools

PCR amplicons were prepared according to the ARTIC-V2 RT-PCR protocol (GunIt) V. 2 with primers for TBEV. Modified primer volumes were used to account for fewer individual primers in the pools and to maintain the same primer concentration in the reaction mixture. PCR amplicon size was checked on 2% agarose gel, and concentration was measured using the Qubit dsDNA HS assay kit on Qubit 3.0 (both Thermo Scientific Inc., Waltham, MA, USA). To reduce the potential interaction between primers in the primer pools, we divided the original two pools into four separate primer pools as indicated in Appendix A.

Further optimization was attempted by combining the primers into nine separate pools to minimize interaction between primers according to the interaction indicated by the Multiple Primer Analyzer online tool (Thermo Scientific Inc.). The optimized primer pools can be found in Appendix A.

### 2.8. Sample Preparation for Shotgun Metagenomic Sequencing

RNA was isolated from blood or serum samples using the EZ1 Virus Mini Kit v2.0 on an EZ1 Advanced XL instrument (Qiagen, Hilden, Germany). DNA was removed using the Turbo DNA-free Kit (Thermo Scientific Inc.) according to the manufacturer’s instructions. cDNA was synthesized with the Maxima H minus double-stranded cDNA synthesis kit (Thermo Fisher Scientific Inc.) using random hexamers. In the second step of cDNA synthesis, we extended the incubation at 16 °C to 2 h. NGS libraries were prepared and sequenced in the same manner as for amplicon sequencing.

### 2.9. Sequencing Using Illumina

NGS libraries of amplicons and double-stranded cDNAs were prepared using the Nextera XT library preparation kit (Illumina, San Diego, CA, USA) according to the product instructions. The concentration of NGS libraries was measured using the Qubit dsDNA HS Assay Kit on the Qubit 3.0 (Thermo Scientific Inc.), and fragment size was analyzed using the Agilent High Sensitivity DNA Kit on the Bioanalyzer 2100 (both Agilent Technologies, Santa Clara, CA, USA). Samples were sequenced using the MiSeq Reagent Kit v3 (600 cycles) on the MiSeq Sequencer or the NextSeq 500/550 High Output Kit v2.5 (300 cycles) on the NextSeq 550.

### 2.10. Sequencing with Oxford Nanopore Technologies

NGS libraries for Oxford Nanopore Technologies (ONT, Oxford, UK) were prepared from purified amplicons according to the protocol for Direct cDNA Sequencing (SQK-DCS-109; version DCS_9090_v109_revN_14Aug2019) starting at the End prep step. Overnight sequencing was performed on two FLO-MIN106 flow cells using the GridION platform.

### 2.11. Bioinformatic Analysis

Reads were trimmed using BBduk, part of the BBTools program package [34]. The quality of the raw and trimmed reads was determined using FastQC [35]. Trimmed reads were mapped to the reference genome with NCBI accession number NC_001672.1 using BWA-MEM [36] with default settings. The mapped reads were further processed using Samtools [37]; they were exported as a bam file that was sorted and mate-flagged, duplicate alignments were marked, and the file was indexed. The depth of coverage was calculated using the Samtools depth function. The consensus sequence was generated using IVAR [38] with the settings “-t 0.5 -q 10 -m 1.”

Raw data generated with ONT were basecalled live by Guppy (High accuracy basecalling, v5.0.17) and quality trimmed to Phred score 9. All reads that passed filtering were mapped to the reference genome with NCBI accession number NC_001672.1 using Minimap2 [39] and default settings. The mapped reads were further processed using Samtools [37]; they were exported as a bam file that was sorted and mate-flagged, duplicate alignments were marked, and the file was indexed. The depth of coverage was calculated using the Samtools depth function. The consensus sequence was generated using IVAR [38] with the settings “-t 0.5 -q 10 -m 1.”

### 2.12. Phylogenetic Tree

Twenty-seven consensus sequences generated in this study and selected reference sequences from the NCBI nucleotide databases were aligned using MUSCLE algorithm v3.8.424 [40] in AliView program [41]. Resulted multiple sequence alignment was used for phylogenetic tree building using program IQ-TREE v 2.1.2 with settings “-m MFP -bb 1000 -bnni -nt AUTO -alrt 1000-abayes” [42,43,44]. Phylogenetic tree was drawn with program FigTree v. 1.4.4 [45]. Selected reference sequences from the NCBI database were TBEV-Eu strain Ljubljana I (JQ654701.1), TBEV-Eu Neudoerfl (U27495.1), TBEV-Eu Hypr (U39292.1), TBEV-Eu 262 (U27491.1), TBEV-FE strain Sofjin-HO (AB062064.1), TBEV-Sib strain Vasilchenko (L40361.3), Omsk hemorrhagic fever virus strain Bogoluvovska (AY193805.1), Langat virus strain TP21 (AF253419.1) and Powassan virus (NC_003687.1)

## 3. Results

### 3.1. Comparison of Shotgun Metagenomic Sequencing and the Amplicon-Based Approach

Clinical samples were obtained from 21 patients with laboratory-confirmed TBE in the first phase of their illness. TBEV was sequenced directly from the specimens, using a shotgun metagenomic sequencing and amplicon-based approach on two different sequencing platforms (Illumina and ONT). We sequenced 20 samples with the amplicon-based approach on the Illumina platform (all successful) and 21 samples with shotgun metagenomic sequencing (five successful). To test amplicon-based approach compatibility to ONT, we tested two samples with the highest viral load (both successful). In total, we generated 27 near-complete TBEV genome assemblies (Table 1).

The nearly complete TBEV genome (>90% of genomic bases) was sequenced from 17 samples using the amplicon-based approach and from 5 samples using the shotgun metagenomic sequencing approach. With the amplicon-based approach, we sequenced between 8.6 × 10^7^ and 1.7 × 10^9^ base pairs per sample (median 2.7 × 10^8^) using the Illumina platform and between 4.8 × 10^9^ and 5.2 × 10^9^ (median 5.2 × 10^9^) using the ONT platform. With shotgun metagenomic sequencing, we sequenced between 3.1 × 10^8^ and 1.9 × 10^11^ base pairs per sample (median 2.8 × 109 base pairs). GC content ranged from 54 to 47% (median 52.5%) for the amplicon-based approach, and from 54 to 48% (median 53.8%) for shotgun metagenomic sequencing. The depth of sequencing was 2.5 × 10^4^ for the amplicon-based approach and 2.9 × 10^5^ for shotgun metagenomic sequencing. The percentage of mapped reads to the reference TBEV genome with NCBI accession number NC_001672.1 ranged from 1.8 to 65.9% (median 37.5%) reads for the amplicon-based approach and from 0.001 to 2.4% reads for shotgun metagenomic sequencing (Figure 1).

### 3.2. Comparison of Consensus Sequences of Nearly Complete Genomes Generated with Two Approaches

Comparison of the consensus complete TBEV genome sequences generated using the amplicon-based approach and unbiased shotgun metagenomic sequencing revealed that the amplicon-based approach is suitable for generating complete genome sequences (Table 2). The sequences generated did not differ from the reference genome NC_001672.1 in length or in GC composition.

### 3.3. Consensus Variants

The reconstructed TBEV genome sequences were screened for mutations at the consensus sequence level in comparison with the reference sequence NC_001672.1. We found between 11 and 250 synonymous single-nucleotide polymorphisms (SNPs; median 220) and between 0 and 40 non-synonymous SNPs (median 28) per genome. We found between 0 and 4 SNPs in the 5′ untranslated region (UTR; median 2) and between 0 and 20 SNPs in the 3′ UTR. Figure 2 shows the number and distinct types of mutations found in TBEV genomes using different approaches and sequencing platforms. Figure 3 shows the arrangement of mutations in the sequenced TBEV genomes. Figure 4 shows phylogenetic relations between consensus sequences from samples and their relations to selected sequences from the NCBI database.

### 3.4. Quasispecies

The 27 near-complete genome assemblies of TBEV generated from 21 clinical samples using different approaches were further screened for the presence of quasispecies. The primary threshold for minimum allele frequency to distinguish quasispecies variation from noise arising from sequencing/amplification errors was set at 0.05, as used for cases with heterogeneous variants and low copy numbers [46]. The amplicon-based approach (20 genomes sequenced on the Illumina platform and 2 on the ONT platform) identified 339 synonymous variants, 404 missense variants, 31 variants in untranslated regions, 525 frameshift variants, 28 new stop codons, and 1 lost start codon. Using shotgun metagenomic sequencing (five genomes sequenced on the Illumina platform), we identified 29 synonymous variants, 62 missense variants, 16 variants in untranslated regions, 34 frameshift variants, no new stop codons, and 1 lost start codon.

The median number of mutations found in the genome using the amplicon-based approach is 12 synonymous variants, 4 missense variants, 1.5 variants in untranslated regions, and 2 frameshift variants. Of the 526 frameshift variants found in the amplicon-based approach, 489 were found in two genomes sequenced using ONT, with a median of 238 frameshift variants per genome (Figure 5).

The median number of mutations found in the genome using shotgun metagenomic sequencing is 8.5 synonymous variants, 16 missense variants, 2.5 variants in untranslated regions, and 9 frameshift variants.

## 4. Discussion

TBEV infection may result in an asymptomatic infection, abortive form of the disease, or CNS inflammation, clinically manifested as meningitis, meningoencephalitis, or meningoencephalomyelitis. The underlying reasons for the different manifestations are largely unknown, with patient age and viral subtype being the most important recognized risk factors for the severity of disease. Because TBEV RNA is present only during the first phase of the disease, when symptoms are nonspecific and patients rarely seek medical attention, genome studies are hampered significantly because samples are rarely available. Therefore, the availability of methods for characterization of the TBEV genome in these unique samples is critical to elucidate the influence of viral determinants on the course of the disease.

The aim of this study was to develop new protocols for sequencing the TBEV genome directly from clinical samples of TBE patients. Of 21 TBE clinical samples, 20 samples were sequenced with an amplicon-based approach on the Illumina platform and yielded 16 nearly complete TBEV genomes (>90% of the genome). Four genomes that did not meet the 90% threshold were from samples with a median viral load of 9.3 × 10^3^ copies per ml. Shotgun metagenomic sequencing was successful in five samples, with a median viral load of 1.1 × 10^5^ copies per ml (Table 1). On the ONT platform we sequenced two samples, 2008-P1-S1 and 2020-P19-S1, with the amplicon-based approach and generated 11,087 bp and 11,138 bp long genomes, covering 89.4% and 76.5% of the complete TBEV sequence, respectively.

At present, only 2 of 280 TBEV sequences longer than 10 kb in the NCBI database have been sequenced directly from clinical specimens without prior virus isolation in the in-vitro system. Both TBEV sequences acquired directly from clinical material are from the brain tissue of fatal TBE cases. The first sequence is the TBEV-Sib genome from Novosibirsk, Russia; it was sequenced in 2013 using Sanger sequencing technology [47]. The second sequence is TBEV-Eu, which was generated in 2015 with Illumina technology, from a human cerebellum sample from a fatal TBE case on Kuutsalo island, Finland [48]. The viral load influences the success of generating a complete or near-complete viral genome with shotgun metagenomic sequencing directly from clinical samples. In our case, only clinical samples with viral loads above 3.8 × 10^4^ copies per ml were successfully sequenced without prior amplicon preparation. In the tissue sample from the fatal case from Finland, the viral load was more than 1 million viral copies per microgram of RNA [48].

Published primer pairs for TBEV-Eu genome amplification are based on large overlapping amplicons (longer than 2 kb) and were designed for sequencing samples with higher viral loads, which usually requires amplification of the virus in cell lines [31]. Because a viral load above 10^6^ copies per ml is rare in the blood of TBEV-infected patients [18], we designed new primers for TBEV with shorter amplicons in the range of 450 bp. We used the “jackhammering” strategy, which is more suitable for low viral loads and partially degraded samples [32]. The primers are designed to generate amplicons covering the entire genome in only two separate PCR tubes. Using our amplification strategy, we were able to recover up to 88% of the TBEV genome directly from the clinical sample with a viral load as low as 887 copies per ml. Similar primers for overlapping amplicons are available for TBEV-FE and TBEV-Sib, but they require separate reactions for each amplicon [49].

Analysis of the consensus genomes revealed a median of 220 synonymous SNPs and a median of 28 non-synonymous SNPs per genome compared to the reference strain NC_001672.1. This is comparable to data reported for TBEV genomes isolated in Hungary between 2011 and 2019 (median 226 synonymous SNPs and 31 non-synonymous SNPs) [50]. Different genes have different numbers of SNPs. The most SNPs were found in the second largest gene—the NS3 gene (length 1862 bp)—724 synonymous SNPs, and 49 non-synonymous SNPs. The normalized number of SNPs per genome size in bp shows that the genes have between 0.18 and 0.68 synonymous SNPs per bp and between 0 and 0.26 non-synonymous SNPs per bp. NS4a has the most synonymous SNPs per bp, at 0.68, and no non-synonymous SNPs. NSP4a plays a role in facilitating the formation of viral replication complexes and in counteracting innate immune responses [51]. The UTRs vary in length: the 5′ UTR is approximately 130 bp long and the 3′ UTR is between 300 and 700 bp long. Both UTRs consist of conserved elements responsible for the cycling of the genome during viral translation, replication, and encapsulation, representing the conserved region and variable region [52]. There is a clear but expected difference between the 5′ UTR with 0.068 SNPs per region bp and the 3′ UTR with 0.385 SNPs per region. The 3′ UTR is more heterogeneous in length and sequence than the 5′ UTR and has a longer variable region [53]. Variation in the 3′ UTR has been associated with differences in virulence [54].

Quasispecies help TBEV switch from a mammal to an arthropod host and vice versa [55], and they play a role in the pathogenic potential of the virus [56]. A higher number of minor variants was detected in samples with higher viral load (>10^6^ viral copies/mL) and, thus, bigger virus populations. A bigger viral population leads to higher heterogeneity in viral populations [57].

Comparison of shotgun metagenomic sequencing and our amplicon-based approach to quasispecies detection showed that shotgun metagenomic sequencing found fewer synonymous SNPs (8.5 vs. 12) but more non-synonymous SNPs (16 vs. four) than the amplicon-based approach. The difference could be due to the difference in copy numbers and ratio between quasispecies. The unequal amplification of different quasispecies sequences can change the ratio after PCR amplification. For example, the ratio 1:10 between two different quasispecies could become the ratio 5:100 after PCR amplification and, thus, be indistinguishable from sequencing errors at the cutoff used for mutation frequencies [58,59]. This effect could be mitigated in the future by using microdrop PCR [60].

Lower coverage reduces sensitivity for quasispecies detection. The rates of quasispecies mutations were 11 for synonymous SNPs and 4 for non-synonymous SNPs per genome. Quasispecies variants are defined by non-fixed (quasispecies) mutations; each viral genome may contain none, one, or all non-fixed quasispecies mutations, or any number in between. This suggests an upper bound of the quasispecies variant pool size of up to 215 possible different virus genome sequences (complete genome haplotypes).

This study has several strengths. It was performed on clinical samples without prior virus isolation in in-vitro systems that could affect the viral genome. We successfully generated 21 new, nearly complete consensus TBEV genome sequences from patient samples, contributing to the knowledge on TBEV genetic variability in Europe, because only 2 nearly complete TBEV sequences have been previously generated directly from clinical samples in the NCBI database. The protocols developed were tested on fresh clinical samples and on clinical samples frozen more than a decade ago, with viral loads ranging from 8.9 × 10^2^ to 1.7 × 10^9^ viral copies per ml. The amplicons generated were developed for two of the most commonly used sequencing platforms: Illumina and ONT. We validated our result with unbiased metagenomic sequencing, which confirmed the obtained consensus sequences.

The main limitation of the study is that the oligonucleotide primer pairs developed are specific for the TBEV-Eu subtype and most likely do not apply to all TBEV subtypes. Their performance for other TBEV subtypes could not be determined because they are not present in Slovenia and clinical samples were not available. Comparing the population structure of quasispecies detected by amplicon-based and metagenomic sequencing, we found that genome amplification leads to more quasispecies detected, but this also changes the population structure of quasispecies.

In conclusion, comparison of the complete consensus sequences generated using the amplicon-based approach and shotgun metagenomic sequencing revealed that the amplicon-based approach is more suitable for generating complete or near-complete consensus genome sequences from clinical samples with different viral loads. The amplicon-based approach allows us to recover most of the untranslated regions of the TBEV genome. Nonetheless, shotgun metagenomic sequencing is invaluable for quasispecies detection because it does not introduce bias to high-copy-number quasispecies. This demonstrates that the newly developed primers for the TBEV genome are useful and can generate a TBEV consensus genome directly from fresh or archived clinical samples.

## Figures and Tables

**Figure 1 viruses-14-01267-f001:**
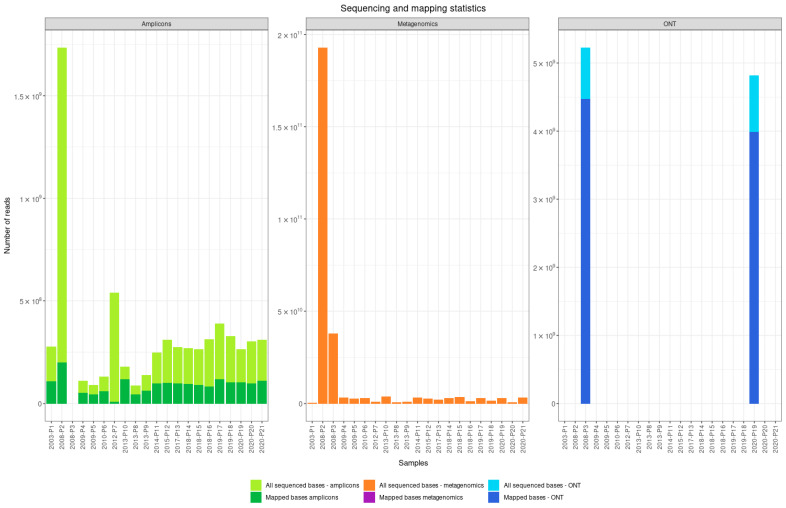
Sequencing and mapping statistics for clinical samples sequenced with amplicon-based and shotgun metagenomic sequencing on Illumina and Oxford Nanopore Technologies platforms. Note the difference in scales on Y axes between the approaches due to difference in number of generated reads.

**Figure 2 viruses-14-01267-f002:**
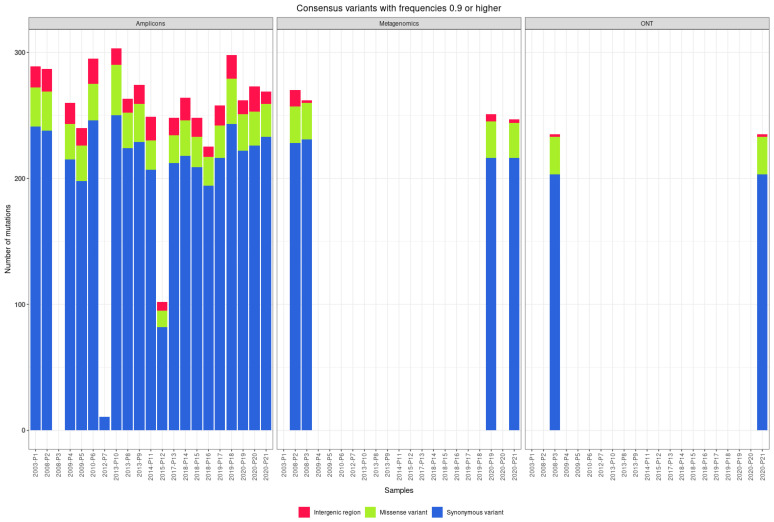
Number and distinct types of mutations found in sequenced TBEV genomes with amplicon-based and shotgun metagenomic sequencing on Illumina and ONT platforms.

**Figure 3 viruses-14-01267-f003:**
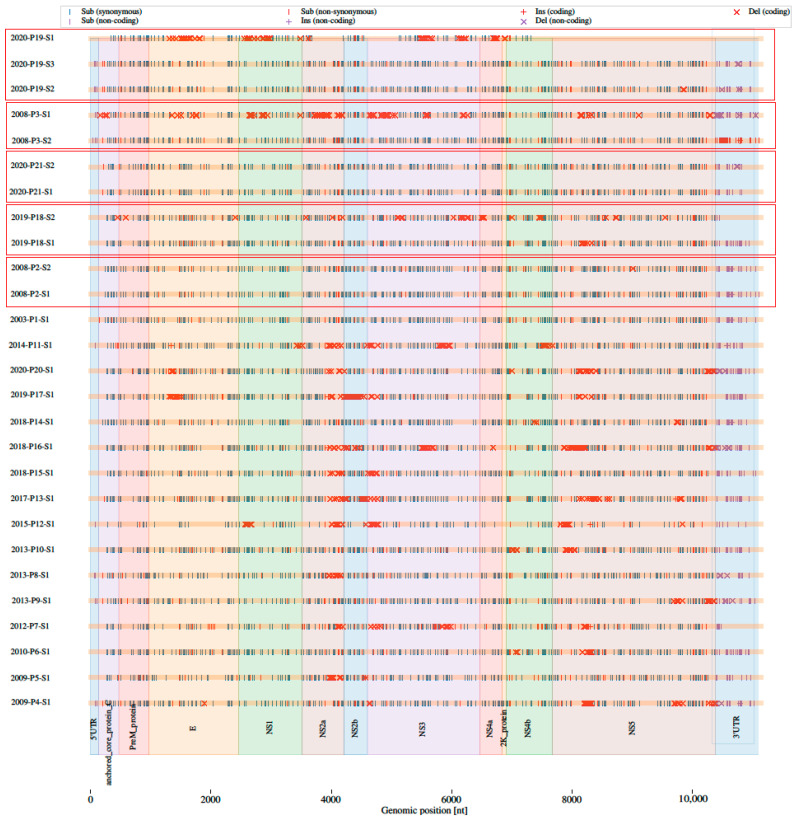
The layout of mutations in sequenced TBEV genomes. Five clinical samples that were sequenced with both approaches and both technologies (shotgun metagenomic sequencing and Oxford Nanopore Technologies) are enclosed in rectangles. Sample ID is composed: year of isolation-patient number–number of an aliquot of the sample used for sequencing.

**Figure 4 viruses-14-01267-f004:**
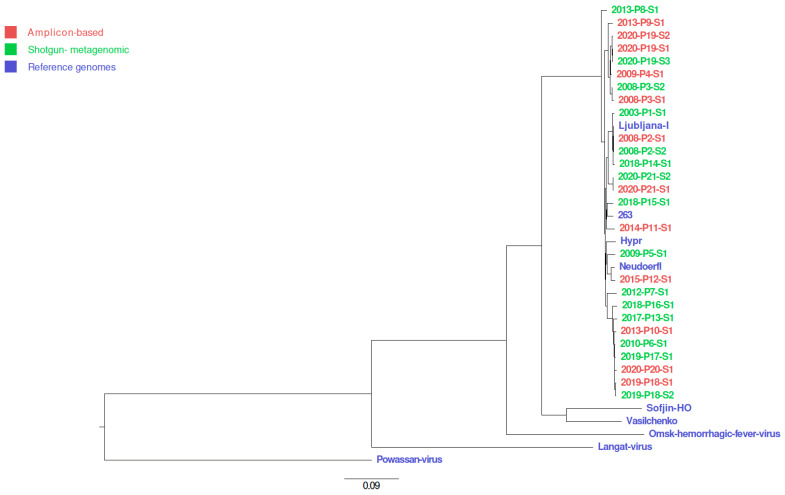
The phylogenetic analysis of generated TBEV consensus sequences. All TBEV genome sequences generated in this study belong to TBEV-Eu subtype. TBEV genome sequences generated from the same clinical sample but with different approaches or sequenced with different platforms cluster together.

**Figure 5 viruses-14-01267-f005:**
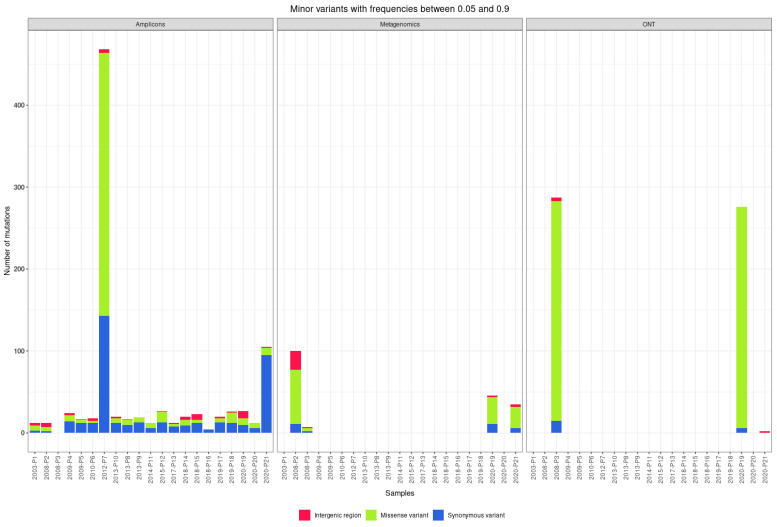
Number of different quasispecies variants found in the sequenced TBEV genomes sequenced with amplicon-based and shotgun metagenomic sequencing on Illumina and Oxford Nanopore Technologies platforms.

**Table 1 viruses-14-01267-t001:** Detailed information on patients and samples included with corresponding viral load and NCBI accession numbers.

Patients	Sample ID	NCBI Accession Number	Year Sample Collected	Sample Type	RNA Number	Viral Load	Approach	Sequencing Platform
Patient 1	2003-P1-S1	ON228409	2003	serum	5171	2.4 × 10^6^	Amplicon	Illumina
Patient 2	2008-P2-S1	ON228408	2008	serum	R1672	1.7 × 10^9^	Amplicon	Illumina
	2008-P2-S2	ON228429					Metagenomic	Illumina
Patient 3	2008-P3-S1	ON228433	2008	serum	R932	4.1 × 10^6^	Amplicon	ONT
	2008-P3-S2	ON228428					Metagenomic	Illumina
Patient 4	2009-P4-S1	ON228422	2009	serum	R2582	5.4 × 10^4^	Amplicon	Illumina
Patient 5	2009-P5-S1	ON228415	2009	serum	R5799	1.1 × 10^5^	Amplicon	Illumina
Patient 6	2010-P6-S1	ON228416	2010	serum	R2758	2.6 × 10^5^	Amplicon	Illumina
Patient 7	2012-P7-S1	ON228425	2012	serum	R4987	6.6 × 10^5^	Amplicon	Illumina
Patient 8	2013-P8-S1	ON228418	2013	serum	R5685	6.6 × 10^5^	Amplicon	Illumina
Patient 9	2013-P9-S1	ON228417	2013	serum	R5837	6.8 × 10^5^	Amplicon	Illumina
Patient 10	2013-P10-S1	ON228419	2013	serum	R5843	4.5 × 10^3^	Amplicon	Illumina
Patient 11	2014-P11-S1	ON228424	2014	blood	NK6645	5.2 × 10^3^	Amplicon	Illumina
Patient 12	2015-P12-S1	ON228423	2015	blood	NK8556	9.4 × 10^3^	Amplicon	Illumina
Patient 13	2017-P13-S1	ON228426	2017	blood	NK8558	8.9 × 10^2^	Amplicon	Illumina
Patient 14	2018-P14-S1	ON228413	2018	blood	NK8563	4.5 × 10^3^	Amplicon	Illumina
Patient 15	2018-P15-S1	ON228421	2018	blood	NK8568	5.6 × 10^3^	Amplicon	Illumina
Patient 16	2018-P16-S1	ON228427	2018	blood	NK2696	3.9 × 10^4^	Amplicon	Illumina
Patient 17	2019-P17-S1	ON228411	2019	blood	1812	5.6 × 10^3^	Amplicon	Illumina
Patient 18	2019-P18-S1	ON228414	2019	blood	NK4357	3.4 × 10^4^	Amplicon	Illumina
	2019-P18-S2	ON228431					Metagenomic	Illumina
Patient 19	2020-P19-S1	ON228434	2020	blood	11599	9.0 × 10^4^	Amplicon	ONT
	2020-P19-S2	ON228412					Amplicon	Illumina
	2020-P19-S3	ON228432					Metagenomic	Illumina
Patient 20	2020-P20-S1	ON228420	2020	blood	12340	5.1 × 10^3^	Amplicon	Illumina
Patient 21	2020-P21-S1	ON228410	2020	blood	NK6108	1.2 × 10^5^	Amplicon	Illumina
	2020-P21-S2	ON228430					Metagenomic	Illumina

**Table 2 viruses-14-01267-t002:** Comparison of consensus sequences generated from the same sample using the amplicon-based or shotgun metagenomic approach and comparison to the reference genome with accession number NC_001672.1. (Abbreviations: aa = amino acids, bp = base pairs, UTR = untranslated region, Ampli = amplicon-based sequencing, Meta = shotgun metagenomic sequencing).

	Length of Sequence		Length of Polyprotein		Length of UTR (5′ URT/3′ UTR) [bp]		GC Content	
	Ampli	Meta	Ampli	Meta	Ampli	Meta	Ampli	Meta
Reference NC_001672.1	11,141		10,245 bp /3414 aa		131/764		54%	
2008-P2-S1/S2	11,142	11,091	10,245 bp /3414 aa	10,245 bp /3414 aa	131/766	131/715	54%	54%
2019-P18-S1/S2	11,009	11,005	10,245 bp /3414 aa	10,245 bp /3414 aa	87/673	87/670	53%	53%
2020-P19-S1/S2	11,008	11,041	10,245 bp /3414 aa	10,245 bp /3414 aa	73/730	106/690	54%	54%
2020-P21-S1/S2	11,009	11,057	10,245 bp /3414 aa	10,245 bp /3414 aa	87/677	111/701	54%	54%

## Data Availability

All generated viral genome sequences acquired in this study were deposited with NCBI. For accession numbers, see Table 1.

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
