# Peer review of "Complete Genome Sequencing of Tick-Borne Encephalitis Virus Directly from Clinical Samples: Comparison of Shotgun Metagenomic and Targeted Amplicon-Based Sequencing"

_viruses, 2022, doi:10.3390/v14061267_

Round 1
Reviewer 1 Report
This paper describes generation and analysis of full-length genomes for TBEV using different NGS techniques (amplicon-based, metagenomics, ONT, Illumina). Although not all samples were comprehensively analyzed with all combinations of ONT/Illumina and Amplicon/shotgun, this study provides sufficient insight and demonstrates useful sequence techniques for TBEV surveillance.
I have a few minor comments:
1) Section 3.2 lines 241-242 mention that no differences were observed between the sequences analyzed and the reference genome in length or in nucleotide composition. In section 3.3 they describe difference between the consensus sequences. I assume that with the first statement the authors mean with composition the %GC and not specific nt changes to the reference genome. To me this was a bit confusing and the authors may want to rephrase section 3.2 if my assumption is correct..
2) Figure 3 is nice but also a bit overwhelming. Have the authors considered to include a phylogenetic analysis and add the phylogenetic tree to the manuscript ? That would give not only a quick indication of differences (or lack there-off) between the methods, but also some idea on the molecular epidemiology of TBEV in Slovenia ?
3) Section 3.4 line 269: The authors set the threshold for noise at 0.05% frequency. Could they further explain why this threshold was chosen ?
4) Could the authors discuss potential reasons why in some samples the number of minor variants is much higher than in others ? Maybe sample type or timing of sampling in relation with virus dynamics, viral load etc.
Author Response
Dear Editor and reviewer,
We would like to thank you for your helpful review of our manuscript “Complete genome sequencing of tick-borne encephalitis virus directly from clinical samples: comparison of shotgun metagenomic and targeted amplicon-based sequencing”. We are thankful for your comments and they have been acknowledged as stated below.
Review Report (Reviewer 1)
This paper describes generation and analysis of full-length genomes for TBEV using different NGS techniques (amplicon-based, metagenomics, ONT, Illumina). Although not all samples were comprehensively analyzed with all combinations of ONT/Illumina and Amplicon/shotgun, this study provides sufficient insight and demonstrates useful sequence techniques for TBEV surveillance.
I have a few minor comments:
- Section 3.2 lines 241-242 mention that no differences were observed between the sequences analyzed and the reference genome in length or in nucleotide composition. In section 3.3 they describe difference between the consensus sequences. I assume that with the first statement the authors mean with composition the % GC and not specific nt changes to the reference genome. To me this was a bit confusing and the authors may want to rephrase section 3.2 if my assumption is correct.
Section 3.2 line 259 was rephrased to clearly mention GC composition.
- Figure 3 is nice but also a bit overwhelming. Have the authors considered to include a phylogenetic analysis and add the phylogenetic tree to the manuscript? That would give not only a quick indication of differences (or lack there-off) between the methods, but also some idea on the molecular epidemiology of TBEV in Slovenia?
According to the reviewer’s suggestion, we have performed the phylogenetic analysis of generated TBEV consensus sequences and selected reference sequences. The TBEV phylogenetic tree was added as Figure 4 in the revised manuscript.
- Section 3.4 line 269: The authors set the threshold for noise at 0.05% frequency. Could they further explain why this threshold was chosen ?
Section 3.4 line 269: the threshold for noise at 0.05 frequency was used as recommended for detecting low and heterogeneous variants and is commonly used. It is used in different fields that study rare variants like oncology and population genetics. Reference was added.
- Could the authors discuss potential reasons why in some samples the number of minor variants is much higher than in others? Maybe sample type or timing of sampling in relation with virus dynamics, viral load etc.
Higher number of minor variants were detected in samples with higher viral load (106 viral copies or more per ml) and thus bigger virus populations. Bigger viral population leads to higher heterogeneity in viral populations. Reference was added.

Reviewer 2 Report
Zakotnik et al. are reporting a novel protocol for obtaining whole TBEV genomes from patient samples. This is a long awaited protocol and will open up excellent opportunities for better understanding if certain TBEV genotypes are associated with different disease severity.
My main comments are three:
- The title says that the paper is about method comparison. However, you mostly report data on targeted amplicon-based sequencing on Illumina. In discussion you actually write that you validated the amplicon-based data by metagenomics instead of comparing these two. Did you do metagenomics for all and only 5 succeeded? Or if you only did 5, how did you choose these? And did you only try 2 on ONT, and if so, why only two and how did you choose these? Please clarify these throughout the manuscript, particularly justifying the choice of the 5 and 2 samples for metagenomics and ONT. And reconsider the title after these modifications.
- The Figures 1,2, and 4 are not clear, please re-think how to present the data. For example in Fig1. you have some light grey color not explained (some samples have up to 8 colors, only 6 have been described?). In Fig2, some samples have 9 colors, only 6 described? And Figure 4 is completely impossible to read, the colors cannot be told apart. Could this be better in a Table format, and maybe as a supplementary one?
- I would have liked to see a phylogenetic tree based on the retrieved whole genomes. I realise the paper is about method development, but still I would have been really eager to see also the benefits of whole genomes in phylogeny. And if there was any differences in phylogeny based on the sequences retrieved by different methods?
Minor comments:
- Introduction, line 60. Is this actually case fatality rate?
- MM, line 117: Do you have a reference for the diagnostic RT-PCR?
- MM 2.8.: How did you choose the samples for metagenomic analysis and how many
- MM 2.10: How did you choose the two samples for ONT
- Figure 3. What are S1, S2 and S3?
Author Response
Dear Editor and reviewer,
We would like to thank you for your helpful review of our manuscript Complete genome sequencing of tick-borne encephalitis virus directly from clinical samples: comparison of shotgun metagenomic and targeted amplicon-based sequencing. We are thankful for your comments and they have been acknowledged as stated below.
Review Report (Reviewer 2)
Zakotnik et al. are reporting a novel protocol for obtaining whole TBEV genomes from patient samples. This is a long awaited protocol and will open up excellent opportunities for better understanding if certain TBEV genotypes are associated with different disease severity.
My main comments are three:
- The title says that the paper is about method comparison. However, you mostly report data on targeted amplicon-based sequencing on Illumina. In discussion you actually write that you validated the amplicon-based data by metagenomics instead of comparing these two. Did you do metagenomics for all and only 5 succeeded? Or if you only did 5, how did you choose these? And did you only try 2 on ONT, and if so, why only two and how did you choose these? Please clarify these throughout the manuscript, particularly justifying the choice of the 5 and 2 samples for metagenomics and ONT. And reconsider the title after these modifications.
All 21 patients’ samples were sequenced with shotgun metagenomic sequencing, but only 5 succeeded, thus we did comparison on five samples. We did amplicon-based sequencing on 20 patient samples (one was left out because of the small available sample volume) and all succeeded. Due to relatively high amount of genetic material required for ONT NGS library preparation and limited sample quantity, only 2 samples with the highest viral load were attempted for ONT sequencing. This was clarified in lines 210-215 in the revised manuscript.
- The Figures 1,2, and 4 are not clear, please re-think how to present the data. For example in Fig1. you have some light grey color not explained (some samples have up to 8 colors, only 6 have been described?). In Fig2, some samples have 9 colors, only 6 described? And Figure 4 is completely impossible to read, the colors cannot be told apart. Could this be better in a Table format, and maybe as a supplementary one?
According to reviewers’ suggestion Figures 1, 2 and 4 were redesigned to be clearer. Different approaches and sequencing platforms were separated and mutations categories were simplified to three main classes – sense, missense and mutations in noncoding regions.
- I would have liked to see a phylogenetic tree based on the retrieved whole genomes. I realize the paper is about method development, but still I would have been really eager to see also the benefits of whole genomes in phylogeny. And if there was any differences in phylogeny based on the sequences retrieved by different methods?
According to the reviewer’s suggestion, a phylogenetic tree was added to the revised manuscript as figure 4. Sequences from the same patient, sequenced with different approach, clusters together.
Minor comments:
- Introduction, line 60. Is this actually case fatality rate?
It is the case fatality rate – corrected in article.
- MM, line 117: Do you have a reference for the diagnostic RT-PCR?
The reference was added to the text.
- MM 2.8.: How did you choose the samples for metagenomic analysis and how many?
According to the reviewer’s suggestion detailed description on samples used for different approaches was added in lines 210-215.
- MM 2.10: How did you choose the two samples for ONT
Clarified in lines 210-215.
- Figure 3. What are S1, S2 and S3?
Designation S1, S2 and S3 represent the sample aliquots, which were used for sequencing with different approaches.